

# Measuring Droplet Fall Speed with a High-Speed Camera:

# Indoor Accuracy and Potential Outdoor Applications

**Cheng-Ku Yu[1], Pei-Rong Hsieh[1], Sandra E. Yuter[2], Lin-Wen Cheng[1], Chia-Lun Tsai[1], Che-Yu Lin[3], and Ying Chen[1]**

[1]Department of Atmospheric Sciences, National Taiwan University, Taipei, Taiwan
[2]Department of Marine, Earth, and Atmospheric Sciences, North Carolina State University, Raleigh, North Carolina, USA
[3]Department of Atmospheric Sciences, Chinese Culture University, Taipei, Taiwan
Correspondence to: C.- K. Yu (yuku@ntu.edu.tw)

**Abstract.** The acquisition of accurate rain drop fall speed measurements outdoors in natural rain represents a long-standing and challenging issue in the meteorological community. Feasibility experiments were conducted to evaluate the indoor accuracy of fall speed measurements made with a high-speed camera and to evaluate its capability for outdoor applications. An indoor experiment operating in calm conditions showed that the high-speed imaging technique can provide fall speed measurements with a mean error of 4.1~9.7% compared to Gunn and Kinzer's empirical fall speed-size relationship for typical sizes of rain and drizzle drops. Results obtained using the same apparatus outside in summer afternoon showers indicated larger, positive and negative velocity deviations compared to the indoor measurements. These observed deviations suggest that ambient flow and turbulence play a role in modifying drop fall speeds which can be quantified with future outdoor high-speed camera measurements. Because the fall speed measurements, as presented in





this article, are analyzed on the basis of tracking individual, specific rain drops, sampling

uncertainties commonly found in the widely adopted optical disdrometers can be significantly

mitigated.

# 1   Introduction

Droplet fall speed (DFS) is an important microphysical parameter playing a key role in modulating

precipitation distributions within three-dimensional storm structures and surface rainfall rates

(Rogers and Yau 1989; Houze 1993; Yu and Cheng 2008; Parodi and Emanuel 2009; Yu and

Cheng 2013). For radar-related and modeling applications, DFS is usually approximated by the

so-called "terminal velocity" ($V_t$), the relative velocity of an object to the air when the aerodynamic

drag force exactly balances the gravitational force. Practically, $V_t$ may be considered to have a

simple one-to-one relationship with raindrop size, and this relationship has been well described in

both theoretical and observational frameworks (Gunn and Kinzer 1949; Atlas et al. 1973; Beard

1976; Doviak and Zrnić 1993). Environmental conditions associated with natural rainfall events

are typically characterized by turbulent air motions and by a population of falling drops with

various sizes. The inertial acceleration of droplets responding to various combinations of wind,

turbulence, collision and breakup may yield appreciable departures of DFS from $V_t$ (Pinsky and

Khain 1996; Pruppacher and Klett 1997; Montero-Martínez et al. 2009). Nevertheless, our

understanding of the degree to which the theoretical value of $V_t$ corresponds to the natural DFS has

been rather limited due to the great challenge of measuring accurate values of DFS in natural

conditions outdoors over a wide spectrum of drop sizes and environments.





In the past century, significant efforts have been made by many researchers to explore a number of different methods for measuring DFS. Lenard (1904) estimated $V_t$ indirectly by measuring the velocity of the air blast by which droplets could be suspended in the air stream. Such suspension techniques were later used to document the behavior of rain drops falling at $V_t$ in many

wind-tunnel studies (Blanchard 1950; Cotton and Gokhale 1967; Pruppacher and Pitter 1971). Other, earlier studies measured DFS by employing shutter and stroboscopic devices (Schmidt 1909; Laws 1941). However, these older investigations had large uncertainties in the measurement of drop size which were estimated using absorbent paper or highly refined flour methods.

A more sophisticated "electronic method" was developed in the 1940's which measured the free

fall speed of artificially generated raindrops inside a laboratory and/or a rain shaft (Wang and Pruppacher 1977). Gunn and Kinzer (1949) (hereafter GK) employed this method to measure $V_t$ in stagnant air by determining the time between the two pulses generated as an electrically charged droplet fell through two inducing rings separated by a known distance. Together with the careful determination of drop size using a weighting method and a microscope, GK were able to provide

accurate measurements of $V_t$ over a wide range of drop sizes. The velocity and size measurements described in GK represent a very reliable reference for theoretical magnitudes of $V_t$ in a standard atmosphere at 20 ℃ and 1013 mb and have been widely used. The methodology employed in GK and similar studies requires a specially designed apparatus operating in a highly controlled environment and is ill-suited for outdoor measurements.

Advancements in photoelectronic techniques since the 1970s have increased the possibilities of measuring DFS associated with natural rainfall events. A growing number of optical instrument



types have been proposed in the literature, such as the spectropluviometer (Donnadieu 1980; Hauser et al. 1984), the Particle Size and Velocity disdrometer (PARSIVEL, Löffler-Mang and Joss 2000), the two-dimensional video disdrometer (2DVD, Schönhuber et al. 1997; Thurai et al. 2013), and the Hydrometeor Velocity and Shape Detector (HVSD, Barthazy et al. 2004). The basic

physical principles underlying these optical instruments are quite similar, with drop size estimated by the degree to which a light sheet is blocked and velocity determined based on the duration of blocking occurrence or the time required to pass through a pair of vertically separate light sheets. Optical disdrometers are designed for outdoor use and can measure simultaneous size and velocity information automatically. Nevertheless, the accuracy of these instrumental measurements is

frequently hampered not only by a variety of sampling uncertainties, such as the splash contamination, margin fallers, and coexistence of two particles inside the light sheet (Löffler-Mang and Joss 2000; Kruger and Krajewski 2002; Yuter et al. 2006; Niu et al. 2010), but also by the assumptions implicit in the algorithms required to automatically determine drop sizes and velocities (Battaglia et al. 2010; Friedrich et al. 2013). In addition, most of these optical

disdrometers cannot distinguish sizes within a size interval (known as the quantization error) and usually suffer from poor signal quality for submillimeter drops (Löffler-Mang and Joss 2000; Yuter et al. 2006). Drop shape deformation and oscillation that usually occur for larger raindrops (diameters > ~1 mm) (e.g., Testik et al. 2006) represent another important uncertainty to the accuracy of these optical disdrometers.

Another group of instruments for retrieving DFS is called the optical array spectrometer probes, as adopted in Montero-Martinez et al. (2009) to study natural DFSs under conditions of weak ambient





winds. Horizontal and vertical extent of the two-dimensional image produced as drops fall past a linear diode array were used to estimate the drop diameter and fall speed, respectively. This methodology involves a theoretical approximation of drop shape deformation for size and velocity determination, as well as possible sampling uncertainties such as those usually found in the aforementioned optical disdrometers. These inherent limitations similarly cause lower precision in the DFS measurement.

An intuitive way of measuring DFS across a wide range of drop sizes is to use a high-speed camera (hereafter HSC) that acquires a set of images of the same particle with adequate spatial and temporal resolution to permit clear identification of its shape and position. A HSC has been used to investigate the behavior of raindrop oscillations and the impact of water drops on the earth's surface (Fukada and Fujiwara 1989; Ghadiri 2006; Testik et al. 2006; Licznar et al. 2008; Szakáll et al. 2010). However, none of these previous published works has addressed the possible application of the HSC to the investigation of atmospheric DFSs. The primary objective of this study is to determine the degree of accuracy of DFS measurements made with a HSC and further to understand its potential for measuring DFSs in the outdoor environment. Photographed images of artificially-created, freely falling water drops of various sizes (diameters from ~0.2 to ~3 mm) from an indoor experimental work were first analyzed to calculate DFS values. The calculated DFS were then compared with theoretical values of $V_t$ to provide quantitative evaluation for the velocity measurement obtained from the HSC. A set of outdoor experiments were also undertaken to evaluate the capability of a HSC to study DFSs associated with natural rainfall events.

## 2   Instruments and indoor experimental settings



The indoor experiment was conducted in the interior staircase of the Ta-Shiao building located within the campus of the Chinese Culture University (CCU), Taipei, in June 2012. The instruments and experimental settings used are illustrated schematically in Fig. 1. The key instrument for the experiment was a HSC with a 400-mm lens focal length, operated with a recording frame rate of 3,600 fps and a shutter speed of 50,000 s$^{-1}$. To improve the spatial resolution of photographed images, two extension tubes were mounted with the lens, resulting in a view frame of approximately $29 \times 29$ mm$^2$ (corresponding to $1,024 \times 1,024$ pixels) and a focal plane at a distance of ~106 cm from the lens. In this setting, the pixel size was quite small, approximately $0.028 \times 0.028$ mm$^2$, allowing for better identification of the outline and shape for the typical sizes of rain and drizzle drops. A portable computer installed with image processing software was connected to the HSC, providing a real-time recording and visualization of the photographed water drops.

In this study, hypodermic needles with various pinhole sizes and the sprinkling method (Magono et al. 1963) were used to generate large (> ~2 mm) and small (< ~2 mm) water drops, respectively. The artificially-created water drops were released at a distance of ~12 m above the camera. This distance is close to the theoretical and experimental prediction of the distance required for large drops (greater than 2 mm) to reach the $V_t$ from rest under atmospheric conditions of 1000 mb and 20 °C (Wang and Pruppacher 1977). The bright-field illumination technique (Cannon 1970; Jones et al. 2003; Testik and Barros 2006), provided by a light source standing in front of the lens (Fig. 1), was used to produce a bright background and a dark drop silhouette. Because of the high recording rate and inherent limitation of storage memory, only a very short duration of ~1.5 s (corresponding to ~5,400 frames) was used for each recorded period. The images obtained during each recorded





period were then checked visually to select particular water drops with a distinct and well-defined shape and outline. Blurred images of water drops that fell outside the narrow focal zone were excluded from this study. The recorded images from a total of 95 water drops in the focal plane with a range of diameters from ~0.2 to ~3 mm were collected for subsequent velocity and size

analysis.

Other instruments employed in the indoor experiment are a PARSIVEL disdrometer and a lightweight Vaisala weather transmitter (WXT520). The PARSIVEL disdrometer was situated between the light source and lens, with its sensing area roughly collocated with the focal plane. The primary purpose of deploying this optical instrument was to provide independent

measurements for initial comparisons with the DFS values measured by the high-speed camera. The thermodynamic and wind conditions within the experimental room were automatically monitored by a WXT520 sensor mounted at a height of ~9 m above the floor. The measurements taken during the collection of the analyzed images indicate a nearly calm condition (a mean wind speed of 0.07 m s$^{-1}$) with average temperature, pressure, and relative humidity equal to 30.2 $^{o}$C,

956.4 mb, and 53%, respectively.

## 3   Determination of drop size and velocity

Under bright-field illumination the photographed water drops appear in the recorded image as a

darker area. Figure 2 shows a sample image of a photographed water drop and its corresponding background image taken just before it fell into the view frame of the lens. There was a sharp transition from light gray to darker gray pixels near the surface of the water drop (Fig. 2a), yielding





a pronounced gradient of brightness values[1] characterizing the outline region. Near the drop center there were also some changes in brightness, related to specular reflection of the light source.

To determine the drop outline, we consider both the brightness difference between the lighter background (Fig. 2b) and the darker drop (Fig. 2a) and the local gradient of brightness. The

brightness gradient was determined using the four-connected pixels in the vertical and horizontal. Both brightness difference and brightness gradient were calculated for each pixel in each image containing a water droplet. The mean gradient values averaged within each interval of brightness difference and plotted as a function of brightness differences are shown in Fig. 3. The analysis indicates that a well-defined threshold value of brightness difference coinciding with the peak

gradient of brightness, presumably marking the drop surface, is approximately equal to 26.

With the brightness characteristics of the drop images described above, two objective methods may be used to determine the drop size. The first method was to calculate the brightness difference between the drop and its background image for each pixel within the view frame and to mark the drop area of the pixels with the threshold of 26 as described above. Once the two-dimensional drop

outline was obtained, the drop volume (*vol*) was calculated with an integration technique by summing the volumes of three-dimensional disks with thickness and diameter corresponding to the height of one pixel and each horizontal pixel row, respectively, as described in Jones and Saylor (2009). The equivalent diameter of the drop (hereafter $D_e$) could be derived directly from the calculated drop volume through the formula $D_e = (6vol/\pi)^{1/3}$. The second method used a procedure

similar to the first, except that the drop outline was adaptively determined by the peak value of

---

[1] The range of the brightness values is from 0 (black) to 255 (white).




brightness gradient found along each radial direction from the drop center.

Figure 4 illustrates the objectively determined drop outlines and their corresponding diameters for three different water drop sizes. For medium ($D_e$=1.9 mm) and large ($D_e$=3.0 mm) water drops, the drop outlines and equivalent diameters determined by the two methods were nearly identical (i.e.,

within 2.5%, Fig. 4c-f). For the small ($D_e$=0.5 mm) water drop, the difference in $D_e$ between the two methods became larger (~15%) (Fig. 4a,b). In particular, the criterion using the radial gradient of brightness value yielded a clear deviation of the drop outline from a spherical shape (Fig. 4b), which is obviously not realistic given the small size of the drop. In fact, experience indicates that this method generally has a larger potential uncertainty in determining the size of small drops

because the brightness contrast across their outline is usually less distinct. In view of this limitation, the criterion based on the single threshold brightness difference between the drop image and its background image was adopted for size determination in this study. For a given water drop, there were a number of photographed images within the view frame and a representative size was then obtained by averaging sizes from all in focus images. It is noteworthy that the method of detecting

drop outline is generally not a key factor to influence the accuracy of size determination. Instead, the pixel size (i.e., image resolution) compared to drop size is more critical for the size determination. Given the pixel size of 0.028 mm, the minimum resolvable length for the drop image, a potential uncertainty for determining each horizontal pixel row of the drop is $\pm 2$ pixels, yielding a range of size error equal to $\pm 0.040$-0.045 mm.

Drop velocity can be measured directly with the HSC by simply tracking the moving water drop within the view frame in a sequence of images. The geometric center of the drop for each of the





instantaneous images was first determined by calculating the mean spatial coordinate of all pixels

constituting the drop. Figure 5 shows a sample plot produced by compositing multiple sequences

of drop images and their corresponding geometric centers. In principle, a drop's velocity can be

calculated by the distance between the geometric centers from two successive or arbitrary drop

images divided by their recorded time difference. However, we consider a specific distance

between the highest and lowest geometric centers of the photographed in focus water drop (i.e., $d$

in Fig. 5) identified within the view frame and the corresponding duration. A mean, representative

drop velocity can be obtained by this calculation. It is noteworthy that the uncertainty of

determining the geometric center of the drop due to the limitation of pixel resolution would mostly

come from the positions of pixels constituting the drop outline instead of those interior pixels of

the drop. Because the geometric center of a drop is determined by a mean spatial coordinate of all

pixels constituting the drop, the potential error in the drop's position may be approximated by

multiplying the pixel size (i.e., 0.028 mm) by the ratio of the number of pixels within the drop

outline and the number of pixels in the area of the entire drop. For the size range of the studied

drops, the ratio ranges from 0.02 to 0.38. This gives a position error of 0.00056~0.01 mm,

corresponding to a velocity error of 0.002~0.036 m s$^{-1}$ when considering the recording frame rate

of 3,600 fps utilized in this study. These velocity errors due to pixel resolution are much smaller

than the velocity uncertainties related to the size determination as will be discussed in section 5.

**4   Theoretical $V_t$**

The accuracy of DFSs measured by the HSC using the indoor experimental setup is evaluated by

comparing with the $V_t$ – size relations of GK. Foote and duToit (1969) approximated the GK's $V_t$



dataset with an $N_{th}$ degree polynomial of the form:

$$V_0(D) = \sum_{j=0}^{N} A_j D^j \qquad (1)$$

where $D$ is the drop diameter (mm) and $A_j$ are constant values determined by using a least-squares

curve fitting technique. We use $N=9$ and $A_j$ values from Table 1 of Foote and duToit (1969), which

yield an approximation with errors of less than 0.5% over the size range 1.2-5.8 mm and 2% over

the size range of 0.1-1.2 mm. Compared to other common empirical approximations of GK's $V_t$

(e.g., Atlas et al. 1973), which have larger velocity discrepancies for small drops (<0.5 mm), the

expressions of Eq. (1) by increasing $N$ can give much higher accuracy over a wide range of drop

sizes.

For the present experiment, DFS measurements were taken at an altitude of ~375 m (mean sea

level, MSL), with a slightly lower air density than that of the standard atmosphere; therefore, some

velocity adjustments are required for the GK dataset due to the effect of air density. Following

Foote and duToit (1969), a mathematical approximation with the correction factor of air density

can be expressed as

$$V_t(D) = V_0(D) \times (\frac{\rho_o}{\rho})^{0.4} \qquad (2)$$

where $\rho_o$ is the air density of the standard atmosphere (~1.2 kg m$^{-3}$) and $\rho$ is the air density at the

level of observation. The air density for each of the analyzed water drops was calculated based on

the Vaisala thermodynamic measurements taken at their corresponding photographed time. The

density exponent of 0.4 in Eq. (2) is currently the most widely accepted value for adjusting sea



level $V_t$ (Atlas et al. 1973; Sangren et al. 1984). The velocity errors from the predictions of Eq. (2) are within 2.5% over the size range of 3.38-5.95 mm (Foote and duToit 1969). Because of a general lack of actual $V_t$ measurements taken at altitudes above sea level, the optimum magnitude of the density exponent in Eq. (2) has been debated and may vary slightly with drop diameters

5  from 0.4 to 0.45 for the size range of the present analysis (Beard 1985). However, this range of the density exponent only produces a minor difference (~0.5%) in velocity adjustment at the experimental altitude and thus could be considered negligible in this study.

## 5  Quantitative comparisons

The size and velocity distribution of 95 analyzed drops are presented in Fig. 6. For comparison, the theoretical $V_t$ curve drawn from Eq. (2) is superposed on Fig. 6. It is clear that the HSC-observed velocities (sizes) for these drops closely follow or are immediately adjacent to the $V_t$ curve. The differences in velocity between the HSC and theoretical values are overall minor and within 0.3 m

15  s$^{-1}$. The specific accuracy of the HSC-observed DFS ($V$) for a given drop with a diameter $D$ may be evaluated by calculating the velocity deviation ($V_d$) from its corresponding theoretical $V_t$ value. This relationship can be expressed as

$$V_d = V - V_t(D) \tag{3}$$

If the HSC is assumed to have a perfect size determination [i.e., $De$ (the equivalent diameter of the drop as described in section 3) is equal to $D$], $V_d$ can be calculated directly from Eq. (3). However, in realistic situations, some errors in the size determination may occur. In this case, determining $V_d$ is not completely straightforward because the value of the theoretical $V_t$ in Eq. (3) would be





somewhat biased by the presence of size errors. To take this uncertainty into account, Eq. (3) may

be rewritten as

$$V_d = V - V_t(D_e + \Delta D) \qquad (4)$$

where $\Delta D$ is the error in the drop size. Because $\Delta D$ is expected to be much less than $D_e$, Eq. (4)

may be further written as

$$V_d = V - V_t(D_e) - \frac{\partial V_t}{\partial D} \times \Delta D \qquad (5)$$

$$V_d = V_e + V_s \qquad (6)$$

$$V_e = V - V_t(D_e) \qquad V_s = -\frac{\partial V_t}{\partial D} \times \Delta D$$

where $V_e$ is the difference between $V$ and the corresponding $V_t$ at $D_e$ and can be calculated directly

from the HSC measurements, and $V_s$ represents the contribution of the size error ($\Delta D$) to the

velocity deviation ($V_d$). It is clear from Eq. (6) that the presence of $\Delta D$, if any, will lead to the

departure of $V_e$ from $V_d$.

The values of $V_e$ calculated for all analyzed water drops and the percentiles with the normalization

of their corresponding terminal velocities are illustrated in Fig. 7. Shading in the figure represents

the range of the velocity uncertainty [i.e., $V_s$ in Eq. (6)] due to the size error of $\Delta D$, $\pm 0.040$-$0.045$

mm, as described in section 3. The range of velocity uncertainty due to size determination

increases with decreasing drop size. As evident in Eq. (5) and Eq. (6), this is a consequence of the

exponential nature of the $V_t$-$D$ theoretical curve with a steeper slope at smaller drop sizes (cf. Fig.

6). The analysis indicates that $V_e$ values are generally small and most of them range from 0.1 to



-0.2 m s$^{-1}$ (Fig. 7a). In addition, except a few of the smaller drops the HSC-observed DFSs tend to be lower than the theoretical $V_t$ values with typical negative $V_e$ of -0.1~-0.2 (Fig. 7a). This consistent trend may suggest a common existence of slightly positive bias in the size determination (i.e., overestimate of corresponding theoretical $V_t$ value).

For $D_e > ~1$ mm, an average magnitude of the $V_e$ percentile is only 1.86 % and the $V_s$ percentile in Eq. (6) is similarly very small (within 0.5-3%) (Fig. 7b). The $V_e$ percentiles tend to increase with decreasing drop size but they are generally close to or inside the envelope of the velocity uncertainty due to size determination (Fig. 7b). The mean magnitudes of $V_e$ for $0.5 < D_e < 1$ mm and $D_e < 0.5$ mm are calculated to be 6.3% and 6.1%, respectively. For $0.5 < D_e < 1$ mm, the

average magnitudes of the upper and lower bounds of the velocity error [i.e., $V_d$ in Eq. (6)] are calculated to be 1.3% and 12.8%, respectively. For $D_e < 0.5$ mm, they are equal to 20.6% and 20.5%, respectively. For all analyzed drops, the mean magnitudes of $V_e$ and the upper and lower bound of the velocity error are calculated to be 4.1%, 5.6% and 9.7%, respectively.

The results above demonstrate that the HSC-observed DFSs are satisfactorily accurate, compared

to current optical disdrometers for measuring DFS with typical velocity errors of ~10-25% (Löffler-Mang and Joss 2000; Barthazy et al. 2004). The comparison between HSC and PARSIVEL size and velocity measurements for 14 of the larger rain drops (D > 1.75 mm) illustrates the quantization of the PARSIVEL measurements (Fig. 8). These drops are selected for presentation because they were simultaneously observed by both the HSC and the PARSIVEL. It

should be noted that in our indoor experiment, the large drops, such as the 14 drops, were released one by one with some time (~10 s) in between, corresponding to each sampling duration of



PARSIVEL. Therefore, when a drop was measured by HSC (i.e., passing through the focal plane) in a certain time, it is practical to check if the drop was also captured by PARSIVEL at that time. For clarity, each drop has been labeled with digits from 1 to 14 in Fig. 8. Within the PARSIVEL sensor precision, the two instruments agree on both size and velocity for 9 out of the 14 drops. For

5    drops 1-4 and 8, the PARSIVEL places the drops into an adjacent size and/or fall speed bin interval to what would be expected based on the more precise HSC measurements. However, the sample size is too small to determine whether these are random or bias errors within the PARSIVEL instrument.

## 6   Outdoor experiments

The capability of investigating DFSs associated with natural rainfall using the HSC was tested outdoors in the open area of the CCU campus during the summer afternoon showers on 15 August

15    2013 and 25 June 2014. Photographic settings adopted in the outdoor experiment were basically similar to those of the indoor experiment. Some waterproof covers were required to protect the HSC and light source from wetting. Owing to the splash problem that usually occurs as precipitation particles hit the waterproof cover, a longer focal distance is basically required for outdoor applications. However, this setting would result in a larger view frame (i.e., larger pixel

20    size) and thus less accuracy of HSC measurements. To retain the pixel resolution and to mitigate the splash problem, a teleconverter and three extension tubes were used, which allow a longer focal plan at a distance of ~4 m from the lens of HSC. In addition to the Vaisala weather transmitter (WXT520), a 3-Axis Ultrasonic Anemometer was employed closely adjacent to the HSC at the

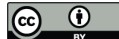



experimental site to provide synchronous high-resolution wind information [sampling rate of 1 (0.1) second for the 2013 (2014) case] with the HSC measurements. The Ultrasonic Anemometer can resolve the three-dimensional wind components in centimeters per second (Siebert and Muschinski 2001). The meteorological conditions during the outdoor experiment were observed to exhibit light

winds, ranging from ~0.3 to ~2.1 m s$^{-1}$, and average temperature, pressure, and relative humidity were equal to 25.5 $^{\circ}$C, 960.6 mb, and 86 %, respectively.

The analysis procedures of determining drop size and velocity for the outdoor experiment generally follow those described in section 3. Because of differences in the degree of indoor and outdoor brightness, the statistical relationship between the brightness difference and the gradient of

brightness (cf. Fig. 3) is also calculated herein, and a threshold value of 35 for the brightness difference is obtained to identify the drop surface for the outdoor experiment. A complication to measurements outside is that the influence of ambient winds causes natural water drops to fall into the view frame of HSC from different angles. For example, a falling drop with motion in a direction perpendicular to the focal plan will feature transition from blurred to clear images (vice

versa) within the view frame. Figure 9 shows a sequence of images photographed as one natural water drop initially was out of the focal zone and then approached and moved into the focal area. In this circumstance, only part of the drop trajectory that is well inside the focal zone (highlighted in Fig. 9) is used for size and velocity calculation.

A total of 29 in-focus natural water drops with different sizes from ~0.2 to ~4.2 mm were collected

during the experiment and their velocity distributions are illustrated in Fig. 10. In contrast to the indoor DFS measurements closely following the theoretical $V_t$ curve (cf. Fig. 6), appreciable





velocity departures of these natural drops from the $V_t$ values are evident. To elaborate whether these velocity deviations are related to the influence of ambient winds and/or turbulences, the horizontal wind speed, vertical velocity, turbulent kinetic energy (TKE), and rainfall rate measured at a time corresponding to each analyzed drop are summarized in Table 1. In the TKE calculation,

the turbulent part is defined as a deviation of measured air velocities from their mean values calculated over a time period of 10 min. The calculated $V_e$ values and percentiles are shown in Fig. 11a and 11b, respectively. In these analyses, each drop has been labeled with digits from 1 to 29 for clarity and discussion.

The analysis reveals that velocity deviations[2] from the theoretical values of $V_t$ vary from drop to

drop range from -1.2 to 0.5 m s$^{-1}$ (Fig. 11a, Table 1). The several drops collected on 25 June 2014 at 15:13:03 UTC when the rain rate was 94.7 mm h$^{-1}$ and the TKE ~1.6 m$^2$ s$^{-2}$ indicate a distribution that includes both positive (drop numbers 2, 5, 11, 14, 21 and 26) and negative/nearly-zero (drop numbers 3, 18, and 29) deviations from the expected values for still air. Montero-Martínez et al. (2009) suggested that when large drops are present super terminal speeds

can occur related to the collision-breakup-relaxation process. The largest deviation from expected value (~30%) is drop 13 which coincided with a rain rate of 0.3 mm h$^{-1}$, TKE =1.9 m$^2$ s$^{-2}$ and a relatively stronger wind speed (~2 m s$^{-1}$). For other drops obtained in conditions of lower TKE, there is also a range of both positive and negative deviations. Our limited data suggest complicated behavior of natural DFSs in the turbulent environment (Pinsky and Khain 1996; Pruppacher and

Klett 1997).

---

[2] Velocity deviations (instead of velocity errors) are stated herein because the theoretical $V_t$ value may not be a perfect ground-truth velocity reference for complicated outdoor environment.





Because the number of our analyzed drops obtained outdoors is limited, we cannot make any firm conclusions regarding the statistical characteristics of natural DFS, and the results presented above may just represent a preliminary assessment of potential outdoor applications for HSC. These initial analyses indicate gaps in our knowledge of how ambient winds and turbulence impact

natural DFS which can be explored with future use of the HSC to collect larger dataset of drop images over a wider spectrum of drop sizes and environmental conditions. It is noteworthy that the reliability of the HSC measurements, like the present study, does not rely on the number of collected drops because they are examined on the basis of tracking individual, specific rain drops (Testik et al. 2006). Various sampling uncertainties commonly found in the widely adopted optical

disdrometers are significantly mitigated in the proposed high-speed imaging technique.

## 7 Conclusions

How to accurately measure droplet fall speed in natural outdoor conditions has been a long-standing and highly challenging issue in the meteorological community. Designs of the past and current measurement techniques of rain drop fall speed outdoors predominantly involve indirect methods and usually suffer from a wide variety of sampling uncertainties and assumptions implicit in the instrumental algorithms required for automatic determination of drop sizes and

velocities. Evaluation of a high speed camera (HSC) setup based on an indoor experiment shows that our high-speed imaging technique can provide accurate fall speed measurements with a mean error of 4.1~9.7% for typical sizes of rain and drizzle drops compared to the Gunn and Kinzer (1949) empirical size-fall speed relationship. Outdoor observations during summer afternoon

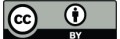

showers demonstrate the capability of investigating natural rain drop fall speeds using the HSC and indicate a potential role for ambient flow and turbulence on contributing to large velocity deviations from the theoretical values of terminal velocity (Pinsky and Khain 1996). Because the HSC measurements, as presented in this article, are analyzed on the basis of tracking individual,

specific rain drops, the application of the proposed HSC technique to the retrieval of fall speed information would not be hampered by various sampling uncertainties and assumptions usually found in the widely adopted optical disdrometers. Future collection of a large dataset of particle images over a wide spectrum of drop sizes and environmental conditions using the HSC will be useful in improving understanding of how ambient winds and turbulence influence natural fall

speeds of rain drops.

## Acknowledgments

The authors would like to thank Chih-Han Peng, Wen-Hsuan Chen, Shiau-Ru Lin, Fang-Ting Li,

Yi-Shan Liao, and Yu-Jen Lien for their assistance with the experimental work. This study was supported by the Ministry of Science and Technology of Taiwan under Research Grant MOST103-2111-M-002-011-MY3. This material is based upon work supported by the United States National Science Foundation under Grant No. AGS 1347491 (Yuter).

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



Table 1. Velocity deviations (i.e., $V_e$) of 29 analyzed drops collected from the outdoor experiments and ambient conditions including horizontal wind speed ($W_S$), vertical air motions ($W_{air}$), turbulent kinetic energy (TKE), and rainfall rate ($R$) corresponding to each analyzed drop.

| Drop No | Time (yymmdd hhmmss) | $D_e$ (mm) | $V_e$ (m s$^{-1}$) | $V_e$ (%) | $W_S$ (m s$^{-1}$) | $W_{air}$ (m s$^{-1}$) | TKE (m$^2$ s$^{-2}$) | $R$ (mm h$^{-1}$) |
|---|---|---|---|---|---|---|---|---|
| 1 | 130815 141645 | 0.2286 | -0.19 | -22.34 | 0.3 | -0.1 | 0.0 | 5.3 |
| 2 | 140625 151303 | 0.2596 | 0.22 | 21.94 | 1.2 | -0.7 | 1.6 | 94.7 |
| 3 | 140625 151303 | 0.3510 | -0.21 | -14.78 | 1.2 | -0.7 | 1.6 | 94.7 |
| 4 | 130815 141238 | 0.4204 | 0.26 | 14.93 | 0.3 | 0.1 | 0.2 | 3.6 |
| 5 | 140625 151303 | 0.4233 | 0.32 | 18.20 | 1.2 | -0.7 | 1.6 | 94.7 |
| 6 | 130815 145817 | 0.6172 | 0.11 | 4.35 | 0.8 | 0.0 | 0.2 | 0.8 |
| 7 | 130815 145511 | 0.7001 | -0.39 | -13.19 | 0.5 | -0.1 | 0.5 | 0.9 |
| 8 | 140625 164225 | 0.7222 | -0.05 | -1.54 | 0.6 | -0.2 | 0.1 | 0.5 |
| 9 | 140625 160243 | 0.7231 | -0.53 | -17.22 | 0.7 | -0.3 | 0.3 | 0.0 |
| 10 | 140625 171849 | 0.8174 | -0.13 | -3.79 | 1.2 | -0.2 | 0.4 | 0.1 |
| 11 | 140625 151303 | 0.8987 | 0.31 | 8.29 | 1.2 | -0.7 | 1.6 | 94.7 |
| 12 | 140625 164225 | 0.9237 | 0.02 | 0.45 | 0.6 | -0.2 | 0.1 | 0.5 |
| 13 | 140625 155949 | 0.9488 | -1.17 | -29.58 | 2.1 | -0.1 | 1.9 | 0.3 |
| 14 | 140625 151303 | 0.9523 | 0.23 | 5.85 | 1.2 | -0.7 | 1.6 | 94.7 |
| 15 | 140625 155637 | 0.9589 | -0.33 | -8.36 | 1.2 | -0.3 | 1.2 | 3.2 |
| 16 | 130815 150104 | 0.9738 | -0.11 | -2.68 | 0.5 | 0.2 | 0.1 | 0.6 |
| 17 | 130815 135706 | 1.1254 | -0.08 | -1.87 | 0.6 | 0.1 | 0.0 | 0.8 |
| 18 | 140625 151303 | 1.1430 | -0.01 | -0.11 | 1.2 | -0.7 | 1.6 | 94.7 |
| 19 | 130815 142202 | 1.2000 | -0.43 | -9.10 | 0.2 | 0.2 | 0.0 | 6.0 |
| 20 | 130815 140332 | 1.2120 | -0.34 | -7.03 | 1.2 | 0.0 | 0.0 | 2.4 |
| 21 | 140625 151303 | 1.2138 | 0.34 | 7.05 | 1.2 | -0.7 | 1.6 | 94.7 |
| 22 | 130815 140332 | 1.2475 | -0.33 | -6.71 | 1.2 | 0.0 | 0.0 | 2.4 |
| 23 | 130815 140818 | 1.5661 | -0.58 | -10.15 | 0.3 | 0.2 | 0.1 | 3.2 |
| 24 | 130815 140818 | 1.9251 | -0.67 | -10.27 | 0.3 | 0.2 | 0.1 | 3.2 |
| 25 | 130815 144018 | 1.9774 | -0.39 | -5.92 | 1.2 | -0.5 | 0.5 | 5.3 |
| 26 | 140625 151303 | 1.9844 | 0.32 | 4.83 | 1.2 | -0.7 | 1.6 | 94.7 |
| 27 | 130815 141645 | 2.1361 | -0.33 | -4.76 | 0.3 | -0.1 | 0.0 | 5.3 |
| 28 | 140625 155222 | 3.3562 | -0.25 | -2.86 | 0.6 | 0.1 | 0.6 | 48.7 |
| 29 | 140625 151303 | 4.2384 | 0.01 | 0.12 | 1.2 | -0.7 | 1.6 | 94.7 |





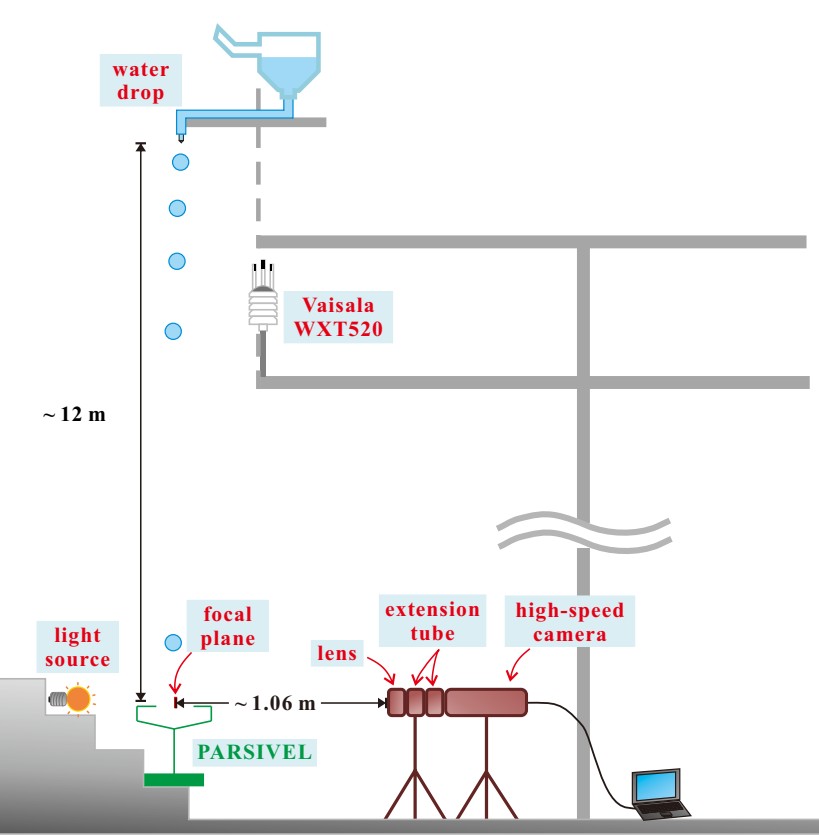

Fig. 1. Instruments and experimental settings adopted for the present study (see text for details).



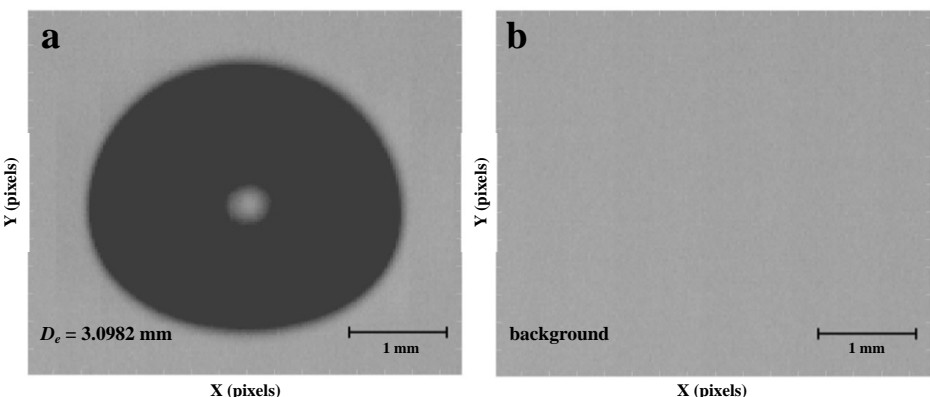

Fig. 2. (a) Sample image of a water drop photographed using high-speed camera. The drop appears as a dark area, and the small, brighter area near the drop center is due to the bright-field illumination adopted in this study. (b) Corresponding background image taken just before the drop fell into the view frame of the lens.



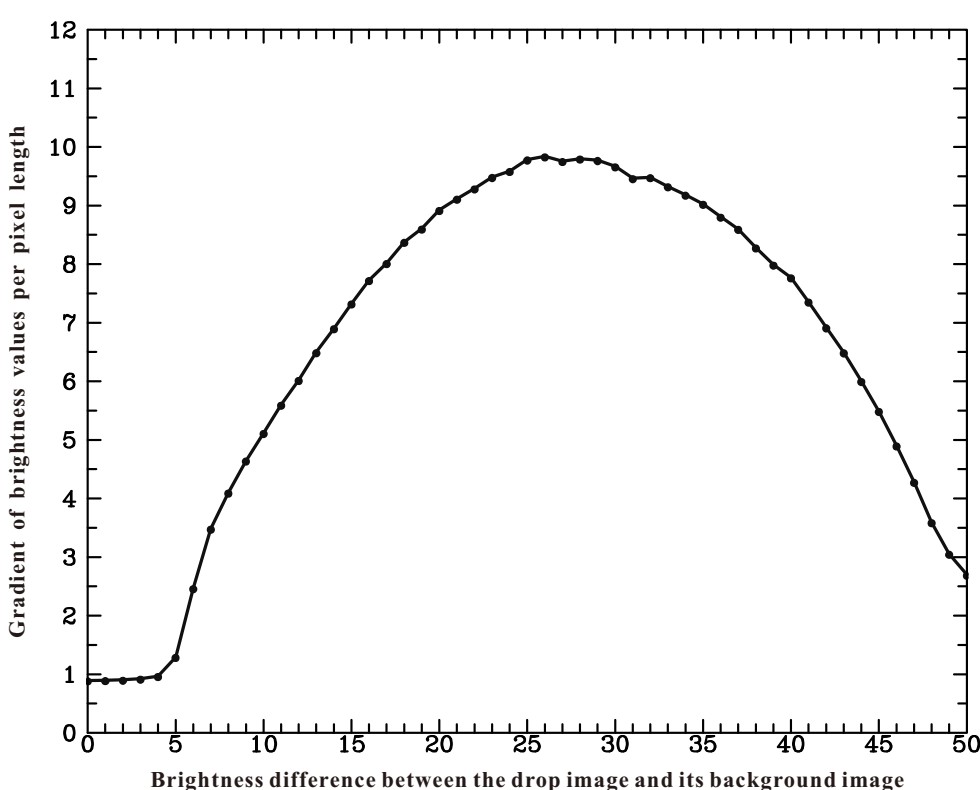

Fig. 3. Statistical relationship between the gradient of brightness values per pixel length (ordinate) and the difference in the brightness values between the drop image and its corresponding background image (abscissa) calculated from all recorded images of 95 water drops collected from the indoor experiment.



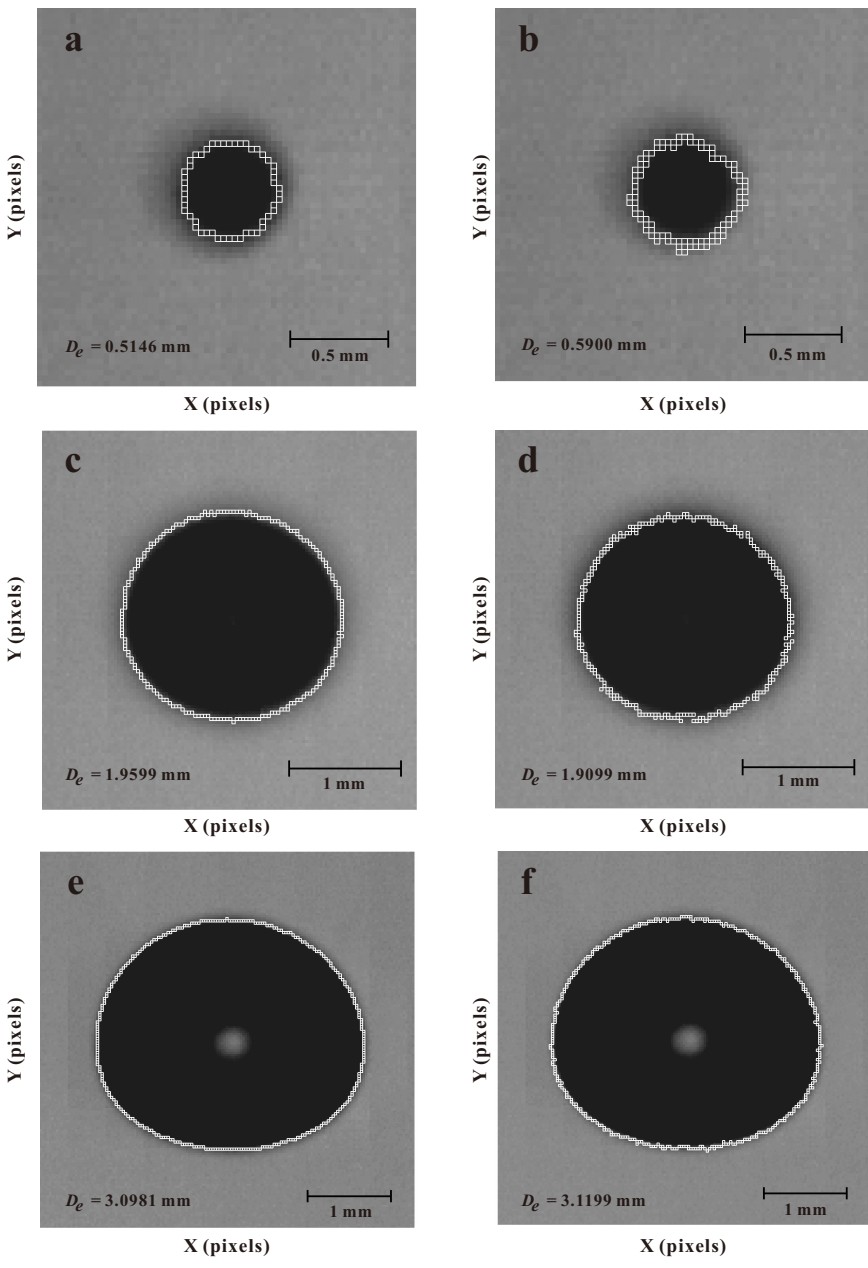

Fig. 4. Drop outlines for three different sizes of water drops (small, medium, and large) determined objectively by the difference in brightness value between the drop image and its background image (a, c, e) and by the gradient of brightness along the radial direction from the drop center (b, d, f). Equivalent diameter of the drop ($D_e$) derived from the determined drop outline is also indicated in each panel.





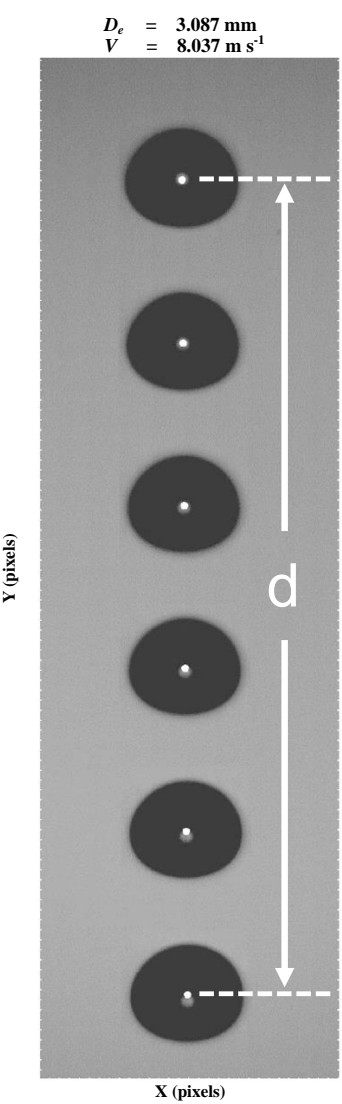

Fig. 5. Multiple sequences of the photographed images as a sample water drop fell into the view frame of HSC. White dots indicate corresponding geometric center of the drop at different time. The vertical distance between the highest and lowest geometric center within the view frame is indicated by "$d$".



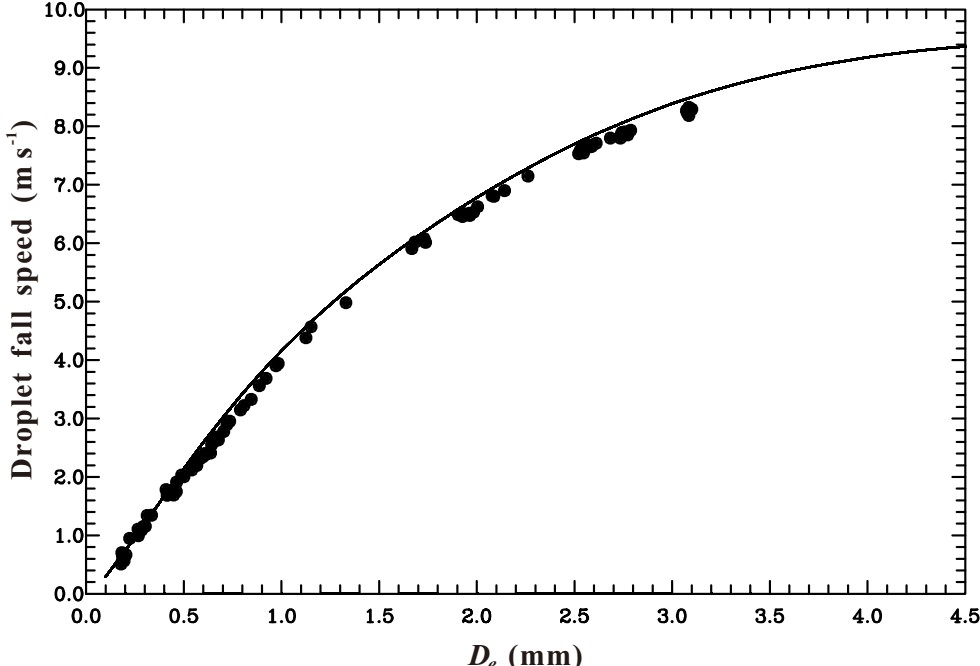

Fig. 6. Size and velocity distribution of 95 analyzed drops collected from the indoor experiment. Each black dot represents a drop. The theoretical curve of $V_t$ is also superposed on the analysis figure.



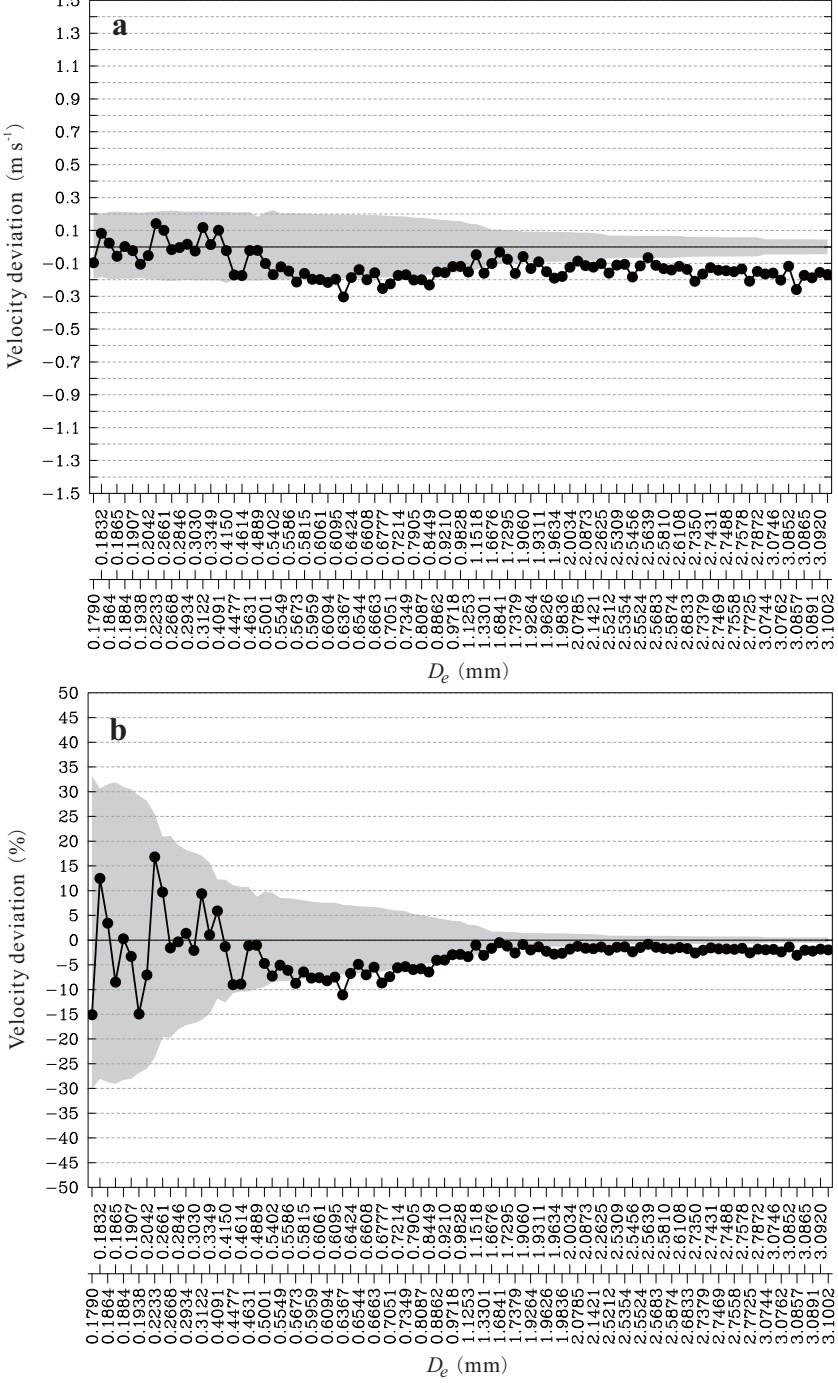

Fig. 7. Velocity error analysis of the HSC-observed DFSs for all analyzed drops. Solid curve with black dot in (a) and (b) indicates respectively the $V_e$ values (i.e., the difference between the HSC-observed DFS and theoretical $V_t$) and their percentiles (i.e., normalized by their corresponding $V_t$ at $D_e$). Shading highlights the range of velocity uncertainties [i.e., $V_s$ in Eq. (6)] due to the potential size error $\Delta D \pm 0.040$-$0.045$ mm associated with the HSC measurements.



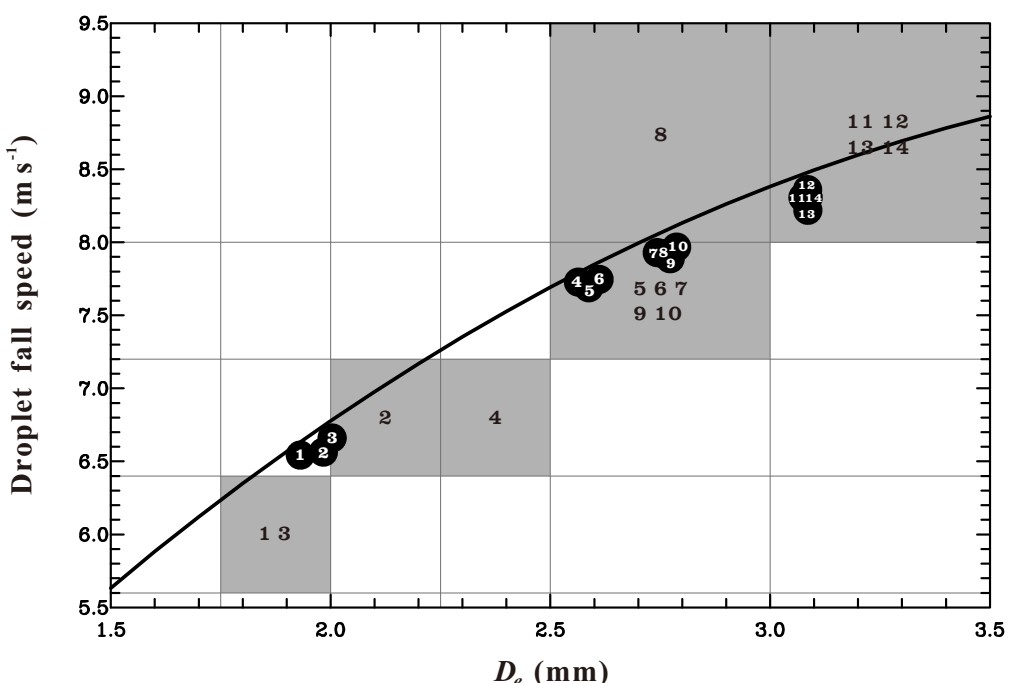

Fig. 8. Size and velocity distribution of fourteen water drops that were observed simultaneously by both HSC and PARSIVEL. Each drop is distinguished with labeled digits from 1 to 14. HSC-observed velocities (sizes) are indicated by solid black circles with white digits and the PARSIVEL measurements are indicated by shading with black digits. The theoretical curve of $V_t$ is also superposed on the analysis figure.



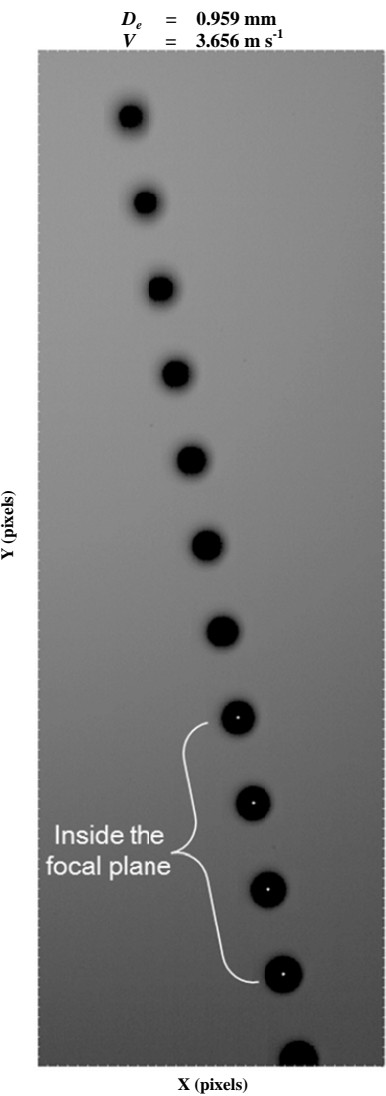

Fig. 9. Multiple sequences of the photographed images for a selected natural water drop ($D_e$ = 0.96 mm) as it was initially out of the focal area with blurred drop outline in the upper portion of the view frame and then moved into the focal zone with sharp and clear drop outline in the lower portion of the view frame.





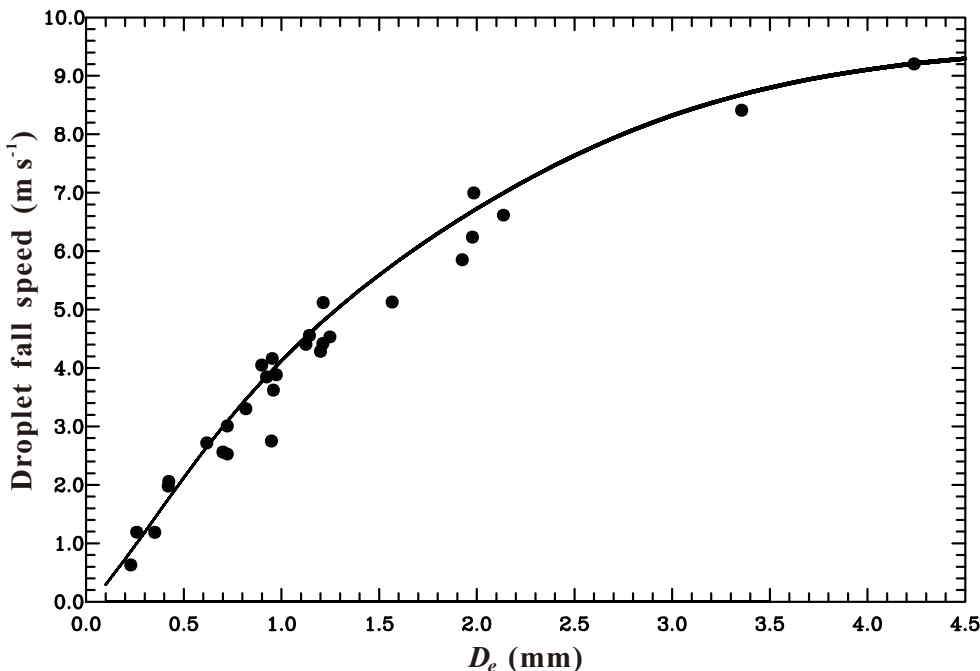

Fig. 10. As in Fig. 6 but showing 29 analyzed drops collected from the outdoor experiments.




Fig. 11. As in Fig. 7 but showing $V_e$ values and percentiles of the HSC-observed DFSs for analyzed natural drops. For clarity and discussion, each drop has been labeled with digits from 1 to 29.