# Peer review of "Measuring Droplet Fall Speed with a High-Speed Camera"

_Atmospheric Measurement Techniques, 2015_

## Referee Comment (RC1) · Anonymous Referee #2 · 3 Feb 2016

General comments

Paper describes most of all results of indoor experiments concerning measurements of droplet fall speed with a High-Speed Camera (HSC) and their accuracy. At the end results obtained for outdoor conditions are also reported and discussed. However presented results for outdoor experiments are limited to 29 drops only collected during 2 storm events. For sure the motivation of study is clear and formulated in following sentence: "The acquisition of accurate rain drop fall speed measurements outdoors in natural rain represents a long-standing and challenging issue in the meteorological community." Despite the rapid progress in electronics and optoelectronics this is still rather a goal to be met than a reality. I would only add that: "Acquisition of accurate

rain drop fall speed measurements outdoors in natural rain by means of moderate cost and easy to use devices represents a long-standing and challenging issue in the meteorological community." I have to also admit that in general the manuscript is well constructed and clearly written. However after manuscript reading I have to raise the fundamental question concerning the novelty of presented study. The detailed literature review of droplet fall speed (DFS) is summarized by the following sentence: "However, none of these previous published works has addressed the possible application of the HSC to the investigation of atmospheric DFSs (page 5, line 13)" I have a doubt concerning the accuracy of this particular statement having in mind references used in manuscript and some other scientific communications. First of all, video disdrometers based on single camera (1 DVD) and double cameras (2 DVD) are already in use and commercially available at JOANNEUM RESEARCH Forschungsgesellschaft mbH. It could be only discussed if this advanced and costly equipment is affordable for all meteorological community. Problems reported in manuscript are already solved in this kind equipment or could be considered as minor in contrast to the issues of fast recorded frames processing, reduction of splash and turbulent wind effects on orifice of devices and supporting optimal cameras and light arrangement for in field measurements. Nevertheless fall velocity, front and side view of every single particle could be acquired by the 2 DVD. Having in mind the journal to which manuscript was submitted I would also suggest to refer to following paper: Garrett, T. J., Fallgatter, C., Shkurko, K., and Howlett, D.: Fall speed measurement and high-resolution multi-angle photography of hydrometeors in free fall, Atmos. Meas. Tech., 5, 2625-2633, doi:10.5194/amt-5-2625-2012, 2012. This particular paper is focused on snowflakes measurements but MASC is based on the idea of HSC image processing, it is capable to measure also other types of hydrometeors and finally some studies of device accuracy are presented. Finally note that in both devices: MASC and 2DVD use of multiple cameras, mounted at different angles solves the problem of focal zone discussed on page 16 and presented in fig. 9.

Specific comments

Page 9, lines 17-19. Why the size error is equal to +/- 0.040-0.045 mm? Should it not be a product of multiplying 2 (or 4, ie. 2 pixels for upper and 2 pixels for bottom droplet edges) by 0.028 mm?

Page 10, lines 15-16. Please check, most probably should be: 0.00056-0.01064 mm and 0.002-0.038 m/s

Page 14, lines 8 12. Most probably instead magnitudes of the Ve - magnitudes of Ve percentile should be used.

Page 16, lines 14-15 – Why not to compare to 2DVD (as claimed by manufacturer: vertical velocity accuracy better than 4

Page 16, lines 16-18 page 15 lines 6-8. Why only larger drops were studied? Parsivel records droplets of diameter smaller than 1.75 mm up to about 0.2 mm.

Page 16, lines 8-11. Does it mean that threshold values could vary between day and night and over the day due to solar radiation differences? If so, this should be commented as another severe complication of outdoor applications.

Page 18 lines 6-8. This sentence is not clear. Note that several drops namely 9 drops were collected simultaneously on 25 June 2014 at 15:13:03 UTC. The question is how many drops simultaneous could we examine especially if a view frame is 29x29 mm2? Could we expect some saturation problems at higher rainfall rates? How much time do we need to process the frames? Is it possible to process them on-line?

Page 18 lines 9-10. Most probably too optimistic having in mind 2 DVD and MASC devices.

Tab. 1. How the rainrate R was estimated? Was it a reading from Parsivel? Please, comment row 9 where you report droplet parameters for rainrate R equal to 0.

Technical corrections

Page 14 line 13 most probably bonds instead bond

Page 15 line 21 most probably allowed instead allow

---

## Referee Comment (RC2) · Anonymous Referee #1 · 4 Feb 2016

General comments:

The accurate knowledge of the terminal velocity of raindrops has high hydrological and meteorological relevance since it is a key microphysical parameter in, e. g., precipitation radar algorithms and precipitation models. It has been a long history of measuring techniques in this field since the beginning of the last century, but there is still a need for precise, accurate, and low cost measuring methods for determining drop fall speeds. The paper of Yu et al. describes an experimental setup utilizing a high speed video camera for terminal velocity measurements. However the setup itself seems to be very simple, there are a lot of difficulties and questions which have to be solved and worked out. The subject of the paper, i.e. the utilization of a relatively new technology for

atmospheric measurements, suits to the scope of AMT and is of high interest to the atmospheric physics community.

In general, the paper is clearly written, well organized and scientifically sounds and can be recommended for publication in Atmospheric Measurement Techniques. Nevertheless, I have some remarks and questions which can be taken into account for a revision before publication:

Specific comments:

line 135-137: The fall distance used in the present setup should also be compared to the results from the very recent paper of Chowdhury et al. (Atmospheric Research, Volume 168, 1 February 2016, Pages 158-168).

line 144: How large is the "narrow focal zone"? If the depth of field is very narrow then the applicability of the HSC will be limited; if it is too large, drop size information will be lost.

line 165: The bright spot inside the drop image cannot be a specular reflection of the light source since it is located on the other side of the object. It is rather a lensing effect.

line 175: From Figure 3 the authors determine an optimal brightness (grey level) value of 26. I would rather say it is 26+/- 2. What error should it cause in the drop size if you use 24 or 28 instead of 26? Further, why didn't you apply the method of Jones and Saylor (2009), where they calculate a histogram for the grey levels and calculate the optimal brightness value?

line 194: The deviation from the spherical shape is realistic. The question is rather how large the axis ratio is, and whether the axis ratio value realistic or not. It should anyway be given a comparison of the axis ratios of the drops to the literature values which could give another approximation of the quality of your size determination.

line 206: I could not follow the estimation for the range of size error. Furthermore, the

drop images shown in Fig.2a/b are very fuzzy, therefore the size error of +/- 2 pixels seems to be unrealistic. A common method for HSC systems is to calibrate the size error of the camera with calibrated spheres, see, e.g., Chowdhury et al. (Atmospheric Research, Volume 168, 1 February 2016, Pages 158-168), please consider to apply it in your study.

line 215: In Figure 5 you indicate the drop size and velocity. It would be desired to know how large d and the corresponding time were.

line 225: Can you provide here in the text an example for a velocity measurement and its error estimation? For example for the drop shown in Fig. 5. It would be easier for the reader to follow your consideration.

line 244-263: Why didn't you use the parameterization of Beard (1976) in which you can set all the relevant physical parameters specifically for your measurement conditions? This parameterization had been proven to work well also for drops with reduced surface tension, for instance (see Müller et al., Atmos Res, 2013).

Fig. 6 and Fig. 10: Please add the error bars to the figures.

Fig. 6. Caption: please refer here to the Equation number for calculating Vt.

line 301: The size error considerations are only valid for rigid drops. But we know that large raindrops are oscillating also in asymmetric modes (see Szakáll et al., 2010, for instance), therefore the integration method may result in false sizes. What error would arise when considering the asymmetrical nature of raindrop shapes after collision, for instance (see Szakáll et al., 2014)? It should also be taken into account or at least mentioned as a source of error.

Fig. 7: Axis labels cannot be read. Furthermore, please indicate in the figure caption what in Fig. 7a and Fig. 7b are plotted.

line 307: The statement holds only if the theoretical values are correct. It would be interesting to see whether the same deviation can be seen when using the parameter-

ization of Beard (1976).

line 348: I guess the focal plane itself was not longer (larger) but its distance to the camera has been increased. Was the depth of field in the outdoor experiments the same as in the indoor ones?

line 371: Here, again, the Beard (1976) parameterization can be applied with the corresponding outdoor parameters.

line 397: How realistic is to collect larger dataset of a rain event? The internal memory of the camera is limited, therefore the saved data should be transferred to a computer. This results in a relatively long idle time, isn't it?

––––––––––––––––––––––––––––

---

## Author Comment (AC1) · 1 Apr 2016

amt-2015-396 Responses to Reviewer#2 1 April 2016

Anonymous Referee #2

Response: We appreciate Reviewer#2's comments, which help us improve the manuscript. A set of responses to your comments is provided below.

General comments Paper describes most of all results of indoor experiments concerning measurements of droplet fall speed with a High-Speed Camera (HSC) and their accuracy. At the end results obtained for outdoor conditions are also reported and discussed. However presented results for outdoor experiments are limited to 29 drops

only collected during 2 storm events. For sure the motivation of study is clear and formulated in following sentence: "The acquisition of accurate rain drop fall speed measurements outdoors in natural rain represents a long-standing and challenging issue in the meteorological community." Despite the rapid progress in electronics and optoelectronics this is still rather a goal to be met than a reality. I would only add that: "Acquisition of accurate rain drop fall speed measurements outdoors in natural rain by means of moderate cost and easy to use devices represents a long-standing and challenging issue in the meteorological community."

Response: In this revision, this sentence (The acquisition of accurate rain drop. . ..) in the abstract has been revised as the reviewer suggested.

I have to also admit that in general the manuscript is well constructed and clearly written. However after manuscript reading I have to raise the fundamental question concerning the novelty of presented study. The detailed literature review of droplet fall speed (DFS) is summarized by the following sentence: "However, none of these previous published works has addressed the possible application of the HSC to the investigation of atmospheric DFSs (page 5, line 13)" I have a doubt concerning the accuracy of this particular statement having in mind references used in manuscript and some other scientific communications. First of all, video disdrometers based on single camera (1 DVD) and double cameras (2 DVD) are already in use and commercially available at JOANNEUM RESEARCH Forschungsgesellschaft mbH. It could be only discussed if this advanced and costly equipment is affordable for all meteorological community. Problems reported in manuscript are already solved in this kind equipment or could be considered as minor in contrast to the issues of fast recorded frames processing, reduction of splash and turbulent wind effects on orifice of devices and supporting optimal cameras and light arrangement for in field measurements. Nevertheless fall velocity, front and side view of every single particle could be acquired by the 2 DVD.

Response: In this revision, the sentence (However, none of these previous. . ..)

has been reworded for clarity. In addition, the authors understand that any instrument/technique has its own strength and weakness. The HSC is not designed nor intended to replace optical distrometers like a PARSIVEL or 2DVD for measurements of fall speeds and drop size distributions. The strength of the proposed HSC method is its high precision without a need of assumptions implicit in the algorithms required to automatically determine drop sizes and velocities, which in turn can effectively mitigate the sampling uncertainties (such as splash contamination, margin fallers, and coexistence of several particles inside the light sheet) if the experimental setups can be appropriately designed. Conversely, a drawback of the HSC technique, as will be explained in our responses below, is related to the time-consuming work during the experimental period. This article represents a feasibility study to understand the specific degree of accuracy of DFS measurements made with a HSC and to evaluate its potential for measuring DFSs in the outdoor environment. Results from the study provide a unique reference in terms of the description of a sensor (i.e., HSC) and its methodology and processing which is a first, necessary step to achieve scientific goals using atmospheric measurements in the future.

Having in mind the journal to which manuscript was submitted I would also suggest to refer to following paper: Garrett, T. J., Fallgatter, C., Shkurko, K., and Howlett, D.: Fall speed measurement and high-resolution multi-angle photography of hydrometeors in free fall, Atmos. Meas. Tech., 5, 2625-2633, doi:10.5194/amt-5- 2625-2012, 2012. This particular paper is focused on snowflakes measurements but MASC is based on the idea of HSC image processing, it is capable to measure also other types of hydrometeors and finally some studies of device accuracy are presented. Finally note that in both devices: MASC and 2DVD use of multiple cameras, mounted at different angles solves the problem of focal zone discussed on page 16 and presented in fig. 9.

Response: Thanks to the reviewer for bringing the article of Garrett et al. (2012) to our attention. In this revision, we have cited this article in the Introduction section and also have emphasized the strength of MASC for solving the problem of focal zone occurring

in our outdoor experiment described in section 6.

Changes in the manuscript: (Introduction) A growing number of optical instrument types have been proposed in the literature, such as the spectropluviometer (Donnadieu 1980; Hauser et al. 1984), the Particle Size and Velocity disdrometer (PARSIVEL, Löffler-Mang and Joss 2000), the two-dimensional video disdrometer (2DVD, Schönhuber et al. 1997; Thurai et al. 2013), the Hydrometeor Velocity and Shape Detector (HVSD, Barthazy et al. 2004), and the Multi-Angle Snowflake Camera (MASC, Garrett et al. 2012).

(Section 6) For example, a falling drop with motion in a direction perpendicular to the focal plan will feature transition from blurred to clear images (vice versa) within the view frame. To solve this problem, multiple cameras with different viewing angles may be deployed in the future, in a manner similar to the instrumental design of the so-called MASC described in Garrett et al. (2012).

Specific comments Page 9, lines 17-19. Why the size error is equal to +/- 0.040-0.045 mm? Should it not be a product of multiplying 2 (or 4, ie. 2 pixels for upper and 2 pixels for bottom droplet edges) by 0.028 mm?

Response: Sorry for the confusing. In this revision, this part of the text has been revised for clarity.

Changes in the manuscript: It is noteworthy that the method of detecting drop outline is generally not a key factor to influence the accuracy of size determination. Instead, relative dimension of the pixel size (i.e., image resolution) and drop size is more critical for the size determination. Given the pixel size of 0.028 mm, the minimum resolvable length for the drop image, it is reasonable to consider a potential uncertainty for determining each horizontal pixel row of the drop equal to 2 pixels. To obtain a maximum (minimum) possible drop size, all of the horizontal pixel rows constituting the drop are increased (decreased) with 2 pixels when integrating the drop volume from each horizontal pixel row. A range of size error may be evaluated by calculating the deviation

of the originally estimated drop size from the calculated maximum/minimum drop size, which is equal to 0.040-0.045 mm.

Page 10, lines 15-16. Please check, most probably should be: 0.00056-0.01064 mm and 0.002-0.038 m/s

Response: Revised as suggested.

Page 14, lines 8 12. Most probably instead magnitudes of the Ve - magnitudes of Ve percentile should be used.

Response: Revised as suggested.

Page 16, lines 14-15 – Why not to compare to 2DVD (as claimed by manufacturer: vertical velocity accuracy better than 4

Response: We agree that the 2DVD fall speed measurements would probably have higher accuracy, compared to other optical disdrometers. However, to our knowledge, the actual degree of the accuracy (or typical errors) of fall speed measured by 2DVD has not been reported in the journal articles or in the formal publications. So, it is difficult for us to make a convincible or direct comparison of their accuracy with the HSC-derived velocity measurements.

Page 16, lines 16-18 page 15 lines 6-8. Why only larger drops were studied? Parsivel records droplets of diameter smaller than 1.75 mm up to about 0.2 mm.

Response: In the indoor experiment, the larger drops, such as the 14 drops selected for the comparison between HSC and PARSIVEL, were generated by using hypodermic needles. These drops were released one by one with some time ($\sim$10 s) in between, corresponding to each sampling duration of PARSIVEL. Therefore, when a drop was captured by HSC (i.e., passing through the focal plane) in a certain time, it is possible for us to check if the drop was also measured by PARSIVEL at that time. For smaller drops (< 2 mm), they were generated by the sprinkling method so it is almost impossible to identify a specific drop captured by HSC from a large population of drops within each

sampling interval of PARSIVEL.

Changes in the manuscript: The comparison between HSC and PARSIVEL size and velocity measurements for 14 of the larger rain drops (D > 1.75 mm) illustrates the quantization of the PARSIVEL measurements (Fig. 8). These drops are selected for presentation because they were simultaneously observed by both the HSC and the PARSIVEL. It should be noted that in our indoor experiment, the larger drops (> 2 mm), such as the 14 drops, were generated by using hypodermic needles as described in section 2. They were released one by one with some time ($\sim$10 s) in between, corresponding to each sampling duration of PARSIVEL. Therefore, when a drop was measured by HSC (i.e., passing through the focal plane) in a certain time, it is practical to check if the drop was also captured by PARSIVEL at that time. For smaller drops (< 2 mm), they were generated by the sprinkling method so it is almost impossible to identify a specific drop captured by HSC from a large population of drops within each sampling interval of PARSIVEL.

Page 16, lines 8-11. Does it mean that threshold values could vary between day and night and over the day due to solar radiation differences? If so, this should be commented as another severe complication of outdoor applications.

Response: In fact, the determination of the drop size is not very sensitive to the threshold of brightness difference we choose. For example, if we use a slightly lower (say 33) or higher (say 37) threshold, it causes a rather minor difference in the drop size (< 1.5 %) compared to that using the original threshold value. This is one of the advantages for the proposed method. In addition, the threshold appears to have a consistent value for a wide spectrum of drop sizes collected from a given experiment. In this revision, we have emphasized this point in section 3.

Changes in the manuscript: It is noteworthy that if we use 24 or 28 as a threshold (cf. Fig. 3), it causes a rather minor difference in the drop size (within 1.5 %) compared to that using the threshold value of 26. The determination of the drop size is not very

sensitive to the threshold we choose.

Page 18 lines 6-8. This sentence is not clear. Note that several drops namely 9 drops were collected simultaneously on 25 June 2014 at 15:13:03 UTC. The question is how many drops simultaneous could we examine especially if a view frame is 29x29 mm2? Could we expect some saturation problems at higher rainfall rates? How much time do we need to process the frames? Is it possible to process them on-line?

Response: In this revision, this sentence has been reworded for clarity. A number of factors including hardware/software settings (the recording frame rate and size of storage memory, for instance) and the characteristics of natural rain (concentration and duration, for instance) can influence the number of raindrops collected by the HSC. In addition, the HSC technique is also related to the time-consuming work during the experimental period, such as the visual and subjective selection of target drops for each recorded period of HSC and the data transfer of these selected drop images from the HSC's temporary storage memory to the hard disk drive of the working computer. These constraints lead to a limited number of water drops that can be actually collected for post-analysis. However, fortunately this weakness can be mostly solved by developing an automatic procedure of judging whether the drops are inside the focal zone and/or by a suitable upgrade in the software/hardware to speed the process of data transfer. These improvements are expected to greatly help strengthen future applications of HSC to the statistical studies of natural DFSs. We are currently undertaking these research and testing works to increase the efficiency of data collection for HSC.

Changes in the manuscript: It is noteworthy that the velocity measurements of HSC, as discussed in this article, are expected to possess good reliability because they are derived on the basis of tracking individual, specific rain drops (Testik et al. 2006).

Page 18 lines 9-10. Most probably too optimistic having in mind 2 DVD and MASC devices.

Response: In this revision, this sentence has been reworded for clarity.

Changes in the manuscript: Various sampling uncertainties can be effectively mitigated in the proposed high-speed imaging technique.

Tab. 1. How the rainrate R was estimated? Was it a reading from Parsivel? Please, comment row 9 where you report droplet parameters for rainrate R equal to 0.

Response: The rain rates shown in Table 1 is provided by the Vaisala weather transmitter (this information has been added in the table caption), not from the reading of PARSIVEL. Because the minimum detection of rainfall intensity for Vaisala is 0.1 mm h-1, it is possible that we can still capture some rain drops during weak rainfall even if the rainfall reading from the Vaisala is equal to zero. This is exactly the case for row 9 indicating a zero rain rate.

Technical corrections Page 14 line 13 most probably bonds instead bond

Response: Corrected as suggested.

Page 15 line 21 most probably allowed instead allow

Response: Corrected as suggested.
* * *

---

## Author Comment (AC2) · 1 Apr 2016

amt-2015-396 Responses to Reviewer#1 1 April 2016

Anonymous Referee #1 General comments: The accurate knowledge of the terminal velocity of raindrops has high hydrological and meteorological relevance since it is a key microphysical parameter in, e. g., precipitation radar algorithms and precipitation models. It has been a long history of measuring techniques in this field since the beginning of the last century, but there is still a need for precise, accurate, and low cost measuring methods for determining drop fall speeds. The paper of Yu et al. describes an experimental setup utilizing a high speed video camera for terminal velocity measurements. However the setup itself seems to be very

simple, there are a lot of difficulties and questions which have to be solved and worked out. The subject of the paper, i.e. the utilization of a relatively new technology for atmospheric measurements, suits to the scope of AMT and is of high interest to the atmospheric physics community. In general, the paper is clearly written, well organized and scientifically sounds and can be recommended for publication in Atmospheric Measurement Techniques. Nevertheless, I have some remarks and questions which can be taken into account for a revision before publication:

Response: We appreciate Reviewer#1's comments, which help us improve the manuscript. A set of responses to your comments is provided below.

Specific comments: line 135-137: The fall distance used in the present setup should also be compared to the results from the very recent paper of Chowdhury et al. (Atmospheric Research, Volume 168, 1 February 2016, Pages 158-168).

Response: Thanks to the reviewer for bringing the published article to our attention. In this revision, we have added a brief discussion for the comparison of our fall distance with the finding from Chowdhury et al. 2016).

Changes in the manuscript: This distance is close to the theoretical and experimental prediction of the distance required for large drops (greater than 2 mm) to reach the Vt from rest under atmospheric conditions of 1000 mb and 20 °C (Wang and Pruppacher 1977). However, the laboratory simulations from a recent study of Chowdhury et al. (2016) have also shown that the required fall distances to reach the Vt are slightly smaller than the theoretical values, with ∼7 (10) m for a drop size of 2.6 (3.7) mm. These results suggest that the fall distance in our experimental setup should be adequate for studying the Vt.

line 144: How large is the "narrow focal zone"? If the depth of field is very narrow then the applicability of the HSC will be limited; if it is too large, drop size information will be lost.

[Figure]

Response: The focal zone is roughly between ∼1 and ∼1.5 cm so it is actually not so narrow. This information has been mentioned in this revision.

Changes in the manuscript: Blurred images of water drops that fell outside the focal zone (1∼1.5 cm) were excluded from this study.

line 165: The bright spot inside the drop image cannot be a specular reflection of the light source since it is located on the other side of the object. It is rather a lensing effect.

Response: We agree. For clarity, we have revised this part of descriptions.

Changes in the manuscript: Near the drop center there were also some changes in brightness, related to the bright-field illumination adopted in this study.

line 175: From Figure 3 the authors determine an optimal brightness (grey level) value of 26. I would rather say it is 26+/- 2. What error should it cause in the drop size if you use 24 or 28 instead of 26? Further, why didn't you apply the method of Jones and Saylor (2009), where they calculate a histogram for the grey levels and calculate the optimal brightness value?

Response: If we use 24 or 28 as a threshold, it causes a rather minor difference in the drop size (within 1.5%) compared to that using the threshold value of 26. Therefore, the determination of the drop size is not very sensitive to the threshold we choose. This is one of the advantages for the proposed method. In addition, the threshold appears to have a consistent value for a wide spectrum of drop sizes. Because the threshold value adopted in Jones and Saylor (2009) was obtained by smoothing the histogram of brightness values for drop images, its uncertainty of determining drop size would be statistically larger, especially for smaller drops (< 1 mm), due to much fewer pixels constituting the drop. This potential drawback was not evaluated in Jones and Saylor, since they consider only larger drops (> ∼1.3 mm) in their experiment.

Changes in the manuscript: It is noteworthy that if we use 24 or 28 as a threshold (cf.

Fig. 3), it causes a rather minor difference in the drop size (within 1.5 %) compared to that using the threshold value of 26. The determination of the drop size is not very sensitive to the threshold we choose.

line 194: The deviation from the spherical shape is realistic. The question is rather how large the axis ratio is, and whether the axis ratio value realistic or not. It should anyway be given a comparison of the axis ratios of the drops to the literature values which could give another approximation of the quality of your size determination.

Response: For the small drop (∼0.5 mm, Fig. 4b) mentioned in this statement, a more spherical shape (i.e., the axis ratio equal to 1) is expected based on theory and observation. However, the irregular distribution of pixels near the surface of the small drop with the criterion using the radial gradient of brightness value is obviously not realistic. This feature is in distinct contrast to a smoother drop surface identified by the brightness difference (Fig. 4a). For clarity, this part of the text has been reworded.

Changes in the manuscript: For the small (De=0.5 mm) water drop, the difference in De between the two methods became larger (∼15%) (Fig. 4a, b). A smoother, reasonable drop surface was obtained with the brightness difference (Fig. 4a). In contrast, the criterion using the radial gradient of brightness value yielded a clear deviation of the drop outline from a spherical shape (Fig. 4b), which is obviously not realistic given the small size of the drop (i.e., ∼0.5 mm).

line 206: I could not follow the estimation for the range of size error. Furthermore, the drop images shown in Fig.2a/b are very fuzzy, therefore the size error of +/- 2 pixels seems to be unrealistic. A common method for HSC systems is to calibrate the size error of the camera with calibrated spheres, see, e.g., Chowdhury et al. (Atmospheric Research, Volume 168, 1 February 2016, Pages 158-168), please consider to apply it in your study.

Response: For clarity, a more detailed description about the estimation for the range of size error has been added in this revision. We agree that the size error of +/- 2 pixels

would be somewhat overestimated or underestimated, but considering all drops we studied, this size error, on average, can be still considered as a reasonable estimate for the uncertainty caused by the limitation of the image resolution. Thanks to the reviewer for bringing the method described in Chowdhury et al. (2016) to our attention. The authors agree that the spherical ball lenses are useful to determine the error of HSC-derived sizes. However, the size error is actually a function of drop size. Also, a wide range of drop sizes from $\sim$0. 2 to $\sim$3 mm has been analyzed in this study. Hence, it is not practical to determine size errors over a wide spectrum of drop sizes using the calibrated spheres.

Changes in the manuscript: It is noteworthy that the method of detecting drop outline is generally not a key factor to influence the accuracy of size determination. Instead, relative dimension of the pixel size (i.e., image resolution) and drop size is more critical for the size determination. Given the pixel size of 0.028 mm, the minimum resolvable length for the drop image, it is reasonable to consider a potential uncertainty for determining each horizontal pixel row of the drop equal to 2 pixels. To obtain a maximum (minimum) possible drop size, all of the horizontal pixel rows constituting the drop are increased (decreased) with 2 pixels when integrating the drop volume from each horizontal pixel row. A range of size error may be evaluated by calculating the deviation of the originally estimated drop size from the calculated maximum/minimum drop size, which is equal to 0.040-0.045 mm.

line 215: In Figure 5 you indicate the drop size and velocity. It would be desired to know how large d and the corresponding time were.

Response: In this revision, the magnitudes of d and the corresponding time have been indicated in Fig. 5.

line 225: Can you provide here in the text an example for a velocity measurement and its error estimation? For example for the drop shown in Fig. 5. It would be easier for the reader to follow your consideration.

[Figure]

Response: In this revision, we have used the drop shown in Fig. 5 as an example to explain our error estimation.

Changes in the manuscript: It is noteworthy that the uncertainty of determining the geometric center of the drop due to the limitation of pixel resolution would mostly come from the positions of pixels constituting the drop outline instead of those interior pixels of the drop. Assuming that all pixels constituting the drop outline have a position error of the pixel size, the potential error in the drop's position may be approximated by multiplying the pixel size (i.e., 0.028 mm) by the ratio of the number of pixels within the drop outline and the number of pixels in the area of the entire drop because the geometric center of a drop is determined by a mean spatial coordinate of all pixels constituting the drop. For example, the ratio and the position error for the larger drop shown in Fig. 5 were calculated to be ∼0.022 and ∼0.0006 mm. With a recording frame rate of 3,600 fps adopted in this study, the position error yields a velocity error of ∼0.002 m s-1. For the size range of the studied drops, the ratio ranges from 0.02 to 0.38. This gives a position error of 0.00056∼0.01 mm, corresponding to a velocity error of 0.002∼0.036 m s-1.

line 244-263: Why didn't you use the parameterization of Beard (1976) in which you can set all the relevant physical parameters specifically for your measurement conditions? This parameterization had been proven to work well also for drops with reduced surface tension, for instance (see Müller et al., Atmos Res, 2013).

Response: Thank the reviewer for the suggestion. The reason why we used Eq. (2) is that its expression is simpler and provides adequate accuracy in the context of our study. We have made comparisons of terminal velocities calculated based on Eq. (2) and that proposed by Beard (1976). The results indicate a rather minor difference, especially for larger drops (> 1 mm) with a velocity difference of 0.06-0.7 %. In this revision, we have cited Beard (1976) herein, along with a brief description, for reader's reference.

[Figure]

Changes in the manuscript: The reason why we used Eq. (2) is that its expression is simpler and provides adequate accuracy. Compared to a more complicated formula of Vt proposed by Beard (1976), there was a very small difference, especially for larger drops (> 1 mm) with a velocity difference of only 0.06-0.7 %.

Fig. 6 and Fig. 10: Please add the error bars to the figures.

Response: The magnitudes of the error bars are typically only a few (tens) cm s-1. Because the range of values (in Y-axis) for these two figures is from 0 to 10 m s-1, the size of the error bars plotted on the figures become very small and is not clearly readable. Since the information about the errors/uncertainties has been indicated in Figs. 7 and 11, we choose to retain the original plot.

Fig. 6. Caption: please refer here to the Equation number for calculating Vt.

Response: Revised as suggested.

line 301: The size error considerations are only valid for rigid drops. But we know that large raindrops are oscillating also in asymmetric modes (see Szakáll et al., 2010, for instance), therefore the integration method may result in false sizes. What error would arise when considering the asymmetrical nature of raindrop shapes after collision, for instance (see Szakáll et al., 2014)? It should also be taken into account or at least mentioned as a source of error.

Response: Careful investigation on the drop images analyzed in this study indicates that there was no obvious evidence of raindrop oscillation. This characteristic is consistent with the fact that the drops captured by the HSC in our indoor experiment have actually reached terminal velocities with equilibrium-shaped status. But, we agree that asymmetric modes of large natural drops due to oscillation and collision can be a possible source of errors influencing the accuracy of the size determination. Thanks to the reviewer for bringing the two published articles to our attention. In this revision, we have citied these works and mentioned the uncertainty associated with the asymmetric

modes

Changes in the manuscript: Shading in the figure represents the range of the velocity uncertainty [i.e., Vs in Eq. (6)] due to the size error of $\Delta$D, 0.040-0.045 mm, as described in section 3. Note that the size error is exclusively related to the limitation of the image resolution and does not consider other sources of errors such as the asymmetric modes of large drops due to oscillation and collision (Szakáll et al. 2010; Szakáll et al. 2014). However, this uncertainty would be relatively minor because the drops captured by the HSC in the indoor experiment is expected to reach terminal velocities with equilibrium-shaped status.

Fig. 7: Axis labels cannot be read. Furthermore, please indicate in the figure caption what in Fig. 7a and Fig. 7b are plotted.

Response: We have rechecked Fig. 7 to ensure clarity.

line 307: The statement holds only if the theoretical values are correct. It would be interesting to see whether the same deviation can be seen when using the parameterization of Beard (1976).

Response: As explained in our response above, the velocity difference in velocity calculated from Eq. (2) and Beard (1976) is very small. The magnitudes of the Ve values and their corresponding percentages based on the formula of Beard (1976) are almost the same as those shown in Fig. 7.

line 348: I guess the focal plane itself was not longer (larger) but its distance to the camera has been increased. Was the depth of field in the outdoor experiments the same as in the indoor ones?

Response: This sentence has been reworded for clarity. Yes, the depth of field is roughly the same as in the indoor ones.

Changes in the manuscript: To retain the pixel resolution and to mitigate the splash problem, a teleconverter and three extension tubes were used, which allow a longer

distance (∼4 m) of the focal zone from the lens of HSC.

line 371: Here, again, the Beard (1976) parameterization can be applied with the corresponding outdoor parameters.

Response: Please refer to our earlier responses concerning the Beard (1976) parameterization.

line 397: How realistic is to collect larger dataset of a rain event? The internal memory of the camera is limited, therefore the saved data should be transferred to a computer. This results in a relatively long idle time, isn't it?

Response: Thank the reviewer for the concern. The HSC technique is related to the time-consuming work during the experimental period, such as the visual and subjective selection of target drops for each recorded period of HSC and the data transfer of these selected drop images from the HSC's temporary storage memory to the hard disk drive of the working computer. These inherent constraints lead to a limited number of water drops that can be actually collected for post-analysis. However, fortunately this weakness can be mostly solved by developing an automatic procedure of judging whether the drops are inside the focal zone and/or by a suitable upgrade in the software/hardware to speed the process of data transfer. These improvements are expected to greatly help strengthen future applications of HSC to the statistical studies of natural DFSs. We are currently undertaking these research and testing works to increase the efficiency of data collection for HSC.